# HLA-E and Its Soluble Form as Indicators of a Sex-Specific Immune Response in Patients with Oral Squamous Cell Carcinoma

**DOI:** 10.3390/ijms242316699

**Published:** 2023-11-24

**Authors:** Anne Radermacher, Michael Fehrenz, Tamara Bellin, Carolina Claßen, Laura Möller, Ann-Kristin Struckmeier, Mathias Wagner, Philipp Wartenberg, Julius Moratin, Christian Freudlsperger, Kolja Freier, Dominik Horn

**Affiliations:** 1Dentistry, Stomatology and Orthodontics, Chair of Oral and Maxillofacial Surgery, Saarland University, 66421 Homburg, Germany; 2Department of Oral and Maxillofacial Surgery, Saarland University, 66421 Homburg, Germany; 3Department of Oral and Maxillofacial Surgery, Friedrich-Alexander-Universität Erlangen-Nürnberg (FAU), 91054 Erlangen, Germany; 4Department of Pathology, Saarland University Medical Center, 66421 Homburg, Germany; 5Department of Experimental and Clinical Pharmacology and Toxicology, Center for Molecular Signaling (PZMS), Saarland University, 66421 Homburg, Germany; 6Department of Oral and Maxillofacial Surgery, University Hospital Heidelberg, 69120 Heidelberg, Germany

**Keywords:** HLA-E, soluble HLA-E, oral squamous cell carcinoma (OSCC), sex-related differences in immune response, immunotherapy

## Abstract

The human leukocyte antigene E (HLA-E) is associated with tumorigenesis in various cancers. Immunoncology along with sex-specific aspects in cancer therapy are now in scientific focus. Therefore, immunohistochemical HLA-E expression was retrospectively analysed in a cohort of oral squamous cell carcinomas (OSCC) after surgical therapy. Then, serum concentration of HLA-E (sHLA-E) was quantified in a prospective cohort by enzyme-linked immunosorbent assay. High HLA-E expression was associated with advanced UICC stage (Spearman’s correlation: *p* = 0.002) and worse survival (Cox-regression: progression-free survival: hazard ratio (HR) 3.129, confidence range (CI) 1.443–6.787, *p* = 0.004; overall survival: HR 2.328, CI 1.071–5.060, *p* = 0.033). The sHLA-E concentration was significantly higher in the control group than in tumor group (Mann–Whitney U-test (MW-U): *p* = 0.021). Within the tumor group, women showed significantly higher sHLA-E levels than men (MW-U: *p* = 0.049). A closer look at the tumor group and the control group showed that gender-specific differences exist: while no differences in sHLA-E concentration were detectable between female subjects of tumor group and control group (MW-U: *p* = 0.916), male subjects of tumor group had a significantly lower sHLA-E concentration compared to those of control group (MW-U: *p* = 0.001). In summary, our results provide evidence for sex-specific differences in immune responses in OSCC. This fact should be considered regarding future immunotherapy regimens.

## 1. Introduction

Head and neck squamous cell carcinoma (HNSCC) is one of the most common cancers worldwide with 377,713 new cases of diseases and 177,757 new cases of death in 2020 [1]. HNSCC as a generic term includes cancers, which mostly derived from the mucosal epithelium cells of the oral cavity, pharynx, and larynx [2]. As one of the most common entities of HNSCC, oral squamous cell carcinoma (OSCC) shows a 5-year overall survival rate of 50–60% [3]. Curative therapy includes surgery in localized disease or the combination with adjuvant radiochemotherapy in locoregional advanced cancer.

Advances in immunotherapy have shifted the palliative regimen from conventional platin-based chemotherapy to the immune checkpoint inhibitors (ICIs). The benefit of these therapeutics was recently approved for OSCC. Pembrolizumab and Nivolumab increase the endogenous immune response to cancer by binding to PD-1 molecules [4]. However, a non-negligible population of the patients do not respond or do not respond durably to a monotherapy with ICIs [5]. The reason for the formation of cellular resistance are immune escape mechanisms of the cancer cells, such as the expression of the non-classical major histocompatibility complex, class I, E (HLA-E) on the cell surface [6].

HLA-E is a surface protein that is physiologically expressed in many different healthy human tissues [7]. It is structured as a heterodimer and consists of a light and a heavy chain, through which it is anchored in the membrane. In contrast to the classical HLA complexes, HLA-E exhibits low polymorphism. In its physiological function, HLA-E mainly presents the leader peptides from classical HLA class-I molecules to immune cells [8,9].

Immune cells such as the natural killer cells (NK) and a subset of the T-cells bind surface HLA-E with high affinity via their inhibitory NKG2A/CD94 receptors. In this way, immune cell activity is reduced by activation of the inhibitory receptors [10]. This mechanism is the target of the therapeutic antibody Monalizumab, which is investigated in ongoing clinical trials [11].

Not only tumor cells use the upregulation of HLA-E as a mechanism to escape the immune response. Also viral-infected cells, for example with HIV or HCV, resort to this type of immune evasion [12,13].

These examples demonstrate that upregulation of HLA-E on tumor or virally infected cells might be associated with a decreased immune response.

Interestingly, this immunological response is not the same in men and women. While women have a higher probability to suffer from autoimmune diseases than men, men have a higher risk of dying from cancer compared to women. At the cellular level, differences in the number of immune cells of the innate and adaptive immune system can be observed between the sexes [14].

Since the immune response is not the same in males and females, the question arises whether there are also sex differences in HLA-E expression or its soluble component, which can be considered an immune-related marker [15]. There are few studies reporting sex-specific differences in HLA levels in different diseases [16,17]. However, the number of studies addressing sex differences in soluble HLA-E concentration in tumor patients is sparse. To our knowledge, a sex-specific difference in HLA-E concentration has not yet been reported in patients with OSCC.

A difference in immune response between men and women with OSCC might suggest a need for sex-specific adjustment of the therapeutic regimen with ICIs.

Therefore, the aim of this study was to retrospectively analyze HLA-E expression in OSCC patients and the impact on tumor progression and survival in a large cohort of patients (n = 222). In a second prospective cohort, the concentration of sHLA-E in the serum of patients with OSCC and a healthy control group was investigated. In addition, we focused on sex differences in sHLA-E levels of men and women with OSCC compared with a healthy control group. Finally, we discussed the results in relation to a sex-specific application of immunotherapeutics.

## 2. Results

### 2.1. Patient Cohort for Tissue Microarray

Overall, 222 patients were analyzed. Of these, 137 were male (61.7%) and 85 were female (38.3%). The age ranged from 28 to 89 years with an average age of 64 years. All tumors were identified as oral squamous cell carcinomas (OSCC) after clinical staging and histopathological examination. The T-classification is distributed as follows: T 1 is represented by 82 (37%) and T 2 by 72 (32.4%) patients. A total of 8 patients showed stage T 3 (3.6%) and 60 patients showed stage T 4 (27%). While 147 subjects showed negative lymph nodes (66.2%), 75 patients had lymph node metastases (33.8%). UICC-Stage I is represented by 69 subjects (31.1%). A total of 44 test persons showed stage II (19.8%) and 23 showed stage III (10.4%). Stage IV is represented by 86 subjects (38.7%). Table 1 shows an overview of the clinical parameters.

### 2.2. Relationship between Clinicopathologic Parameters and HLA-E Expression in the Tissue Microarray Cohort

Older patients (>64 years) showed higher HLA-E expression (Spearman’s rank Correlation (SRC), *p* = 0.007). Also, high T-classification (SRC, *p* = 0.003), high UICC-Stage (Spearman Correlation *p* = 0.002), and the appearance of relapses (SRC, *p* = 0.041) are correlated with higher HLA-E expression. Soft palate and buccal plane tumors showed a higher HLA-E-Expression in contrast to other locations (SRC, *p* = 0.007). Table 2 shows the absolute and percentage distribution of the clinical parameters separated by the HLA-E expression Level supplemented by Spearman’s *p*-value.

### 2.3. Survival Analysis in Relation to the HLA-E-Expression

Regarding Figure 1 and Figure 2, the survival analyses show a significantly reduced overall survival (OS, Log-Rank test, *p* = 0.01, Figure 1) and progression-free survival (PFS, Log-Rank test, *p* = 0.003, Figure 2) for patients with a high HLA-E expression.

### 2.4. Multivariate Analysis

The Kaplan–Meier Curve as a descriptive univariate method of performing survival analysis showed significantly reduced overall survival (OS) and progression-free survival (PFS) for patients with high HLA-E surface expression. As confounders remain unconsidered, a multivariate Cox-regression was performed.

Multivariate Cox-Regression (Table 3) was performed with the clinical parameters Sex, Age, N-classification, UICC-Stage and HLA-E-Expression. The analysis affirmed HLA-E as independent prognostic marker in relation to PFS (HR 3.129 (1.443–6.787), *p* = 0.004) and to OS (HR 2.328 (1.071–5.06), *p* = 0.033), as well as the N-classification (PFS: HR 4.764 (1.607–14.120), *p* = 0.005; OS: HR 2.663 (1.086–6.526), *p* = 0.032).

### 2.5. Clinical Parameters of the Prospective Patient Cohort (OSCC Patient and Healthy Controls) for Serum HLA-E Analysis

In total, the cohort consists of 32 patients with OSCC and 42 healthy members of the control group. The control group consisted of 22 male (52.4%) and 20 female (47.6%) members, 13 were older than 65 years (31%) and 28 members were younger (66.7%). The patient group is composed of 18 male (56.2%) and 14 female (43.8%) participants. A total of 13 of them are older than 65 years (40.6%) and 16 are younger (50%) (Table 4). The mean age in the tumor group is 65 years. The age range extends from 48 years to 93 years (Appendix A). In the healthy control group, the mean age is 59 years. The age interval ranges from 20 to 86 years (Appendix A).

In the tumor cohort, 22 (68.8%) patients had positive HLA-E values in the blood serum tested. In 10 (31.3%) tumor patients no HLA-E could be detected in the serum. In the control group, HLA-E was detectable in 35 (83.3%) subjects. In 7 (16.7%) subjects of the control group no HLA-E could be found.

All tumors were histopathologically classified as squamous cell carcinomas and were located in the oral cavity. T classification was distributed as follows: T1 was diagnosed in 6 (18.8%) and T2 and T3 were diagnosed in 7 patients (T2: 21.9% and T3: 21.9%, respectively). A total of 12 patients had stage T4 (37.4%). 22 subjects in the patient group had negative lymph nodes (68.8%). Lymph node metastases (28.1%) were diagnosed in 9 subjects. UICC stage I was present in 6 patients (18.8%). A total of 5 subjects showed stage II and stage III (UICC II: 15.6%; UICC III: 21.9%) were classified in 7 subjects. Stage IV was diagnosed in 13 patients (40.6%). Table 4 shows an overview of the distribution of clinicopathologic parameters.

Since both the patient group and control group have a small number of participants, normal distribution of the data was first tested with the Shapiro–Wilk (SW) and Kolmogorov–Smirnov (KS) tests (Appendix A). The test results confirmed a lack of normal distribution of the data of sex and HLA-E concentration (HLA-E: KS *p* = 0.02; SW *p* < 0.001).

### 2.6. Correlation between Soluble HLA-E-Concentration (sHLA-E) and the Clinical Parameters in OSCC and Healthy Controls

Enzyme-linked Immunosorbent Assay (ELISA) was performed to investigate the soluble HLA-E level of patients with OSCC. Then, the clinical parameters were correlated with the soluble HLA-E levels. A correlation between HLA-E-level and the sex (Spearman’s rho = 0.357, *p*-value = 0.045) was determined (Appendix A). All other clinical parameters showed no correlation with the soluble HLA-E-levels.

#### Serum sHLA-E Concentration in Tumor Group and in Control Group

To further investigate the significant positive correlation of sHLA-E concentration with sex shown in Spearman’s correlation (Appendix A), a Mann–Whitney U-test was performed. The sHLA-E concentration in blood serum differentiated by sex in the tumor cohort was examined. Comparison of sHLA-E concentration between males and females in this group showed that females in the tumor group had a significantly higher mean rank (20.18) than males (13.64) (Mann–Whitney U-test: N: 32, U = 177.5, W = 282.5, *p* = 0.049) (Appendix A, Figure 3B). No statistical difference could be shown for the sHLA-E concentration between the male and female participants of the control group (Mann–Whitney U-test: N: 42, U = 198.5, W = 408.5, *p* = 0.587) (Appendix A, Figure 3C).

In a further step, the sHLA-E concentration of the tumor group was compared with the sHLA-E concentration of the control group, for which another Mann–Whitney U-test was performed (Appendix A, Figure 3A). The test showed that the sHLA-E concentration in the control group was significantly higher than the sHLA-E concentrations of the tumor group (N = 74, U = 461, W = 989, *p* = 0.021). A closer look reveals that there is a gender difference in sHLA-E concentration between males and females: While in men a significantly lower sHLA-E concentration is detectable in the serum of the tumor group than in the control group (Mann–Whitney U-test: N = 40, U = 312.5, W = 565.5, *p* = 0.001) (Appendix A, Figure 3E), in women no difference in sHLA-E concentration between tumor and control group could be detected (Mann–Whitney U-test: N = 34, U = 134, W = 353, *p* = 0.916) (Appendix A, Figure 3D).

## 3. Discussion

HLA-E has been associated with the genesis of various tumor types in recent years. For some time now, its resolved form, sHLA-E, has also been studied more closely in this context. However, in times of increasing focus on sex-specific medicine, more detailed studies on sex differences in HLA-E expression and sHLA-E levels are lacking. The aim of this study is to investigate the sex differences in HLA-E expression and its solute component in relation to clinicopathological parameters and survival rates in patients with oral squamous cell carcinoma (OSCC).

Our results show that HLA-E expression negatively affects the progression and survival of OSCC patients regardless of sex (UICC: SRC, *p* = 0.002, Multivariate analysis OS: HR = 2.328 *p* = 0.033). In agreement with our results, Babay et al. reported that increased HLA-E expression is associated with advanced disease stage in ovarian cancer [18]. Consistent with other studies, high HLA-E expression was associated with worse survival in this study [19,20]. However, there are also contradictory results that found no association between HLA-E expression and clinicopathologic parameters [21,22,23]. Contradictory to our study, Xu et al. evaluated high HLA-E expression as a favorable prognostic factor in patients with squamous cell carcinoma of the esophagus [24].

One possible reason for the lack of association between HLA-E expression and clinicopathologic parameters in the studies mentioned could be tumor stage: the ovarian and cervical carcinomas studied by Gooden et al. showed an early T stage. It is therefore possible that due to an earlier T-stage a correlation between HLA-E expression and T-stage is less clearly detectable than in tumors with an advanced T-stage [21]. With regard to esophageal cancer, Xu et al. explained the positive impact of high HLA-E expression on survival they found by the complex function of HLA-E in the immune response: HLA-E can have both inhibitory and activating effects on natural killer (NK) cells. Xu et al. deduced the divergent prognostic significance of high HLA-E expression from a possible different effect of HLA-E in different tumors [24].

However, our results show that high HLA-E expression is associated with poorer survival in OSCC patients. This can be explained by the fact that in these tumors the immunosuppressive effect of HLA-E on NK cells seems to predominate. The effect of Monalizumab, which is in the clinical trial phase for the therapy of patients with advanced OSCC and which leads to an activation of the immune system by inhibiting the CD94/NKG2A axis, also supports this assumption [11]. Also, the results of our multivariate analysis show, that HLA-E can be valued as an independent negative prognostic marker for survival of patients with OSCC. 

In the second part of the study, the concentration of the soluble form of HLA-E (sHLA-E) in the blood of a smaller collective consisting of 32 patients with oral squamous cell carcinoma was investigated by ELISA and compared with the sHLA-E concentration of a healthy control group consisting of 42 subjects. The aim was to investigate potential sHLA-E differences between the tumor and healthy control groups, especially against a gender-differentiated background.

It was shown that the sHLA-E concentration of the control group was significantly higher than the sHLA-E concentration of the tumor group (Mann–Whitney U-test: N = 74, U = 461, W = 989, *p* = 0.021). Closer inspection revealed that there was a gender difference in sHLA-E concentration between men and women: while in men a significantly lower sHLA-E concentration was detected in the serum of the tumor group compared to the control group (Mann–Whitney U-test: N = 40, U = 312.5, W = 565.5, *p* = 0.001), in women no statistically significant difference in sHLA-E concentration between tumor group and control group could be detected (Mann–Whitney U-test: N = 34, U = 134, W = 353, *p* = 0.916).

The current literature comparing sHLA-E differences in the blood of tumor patients with a healthy control group is sparse. Allard et al. were able to demonstrate a significantly higher sHLA-E concentration in the blood serum of melanoma patients than in the healthy control group [15]. In further studies, the sHLA-E concentration was also higher in the tumor group than in the control group [17,25]. However, not all studies demonstrate increased sHLA-E concentration in blood serum: for a cohort consisting of neuroblastoma patients, Morandi et al. did not demonstrate an increased sHLA-E concentration in blood serum compared with a healthy control group [26]. They conclude that the involvement of neuroblastoma cells in sHLA-E release is unlikely. A similar issue was observed in the present study: no difference could be found between the sHLA-E concentration of female tumor patients in the healthy control group.

While the results of the TMA cohort represent the immunological situation of the solid tumor, the results of the ELISA cohort allow insight into the effect of the tumor on the whole system. Thus, the results of the TMA cohort show a positive correlation between HLA-E expression and clinicopathological parameters, which could no longer be detected in the ELISA cohort. On the other hand, the ELISA cohort showed a significant sex-specific difference in sHLA-E concentration between the tumor group and the control group. The question therefore arises whether there must necessarily be a correlation between surface HLA-E expression and blood sHLA-E concentration.

For a cohort consisting of thyroid carcinoma patients, Kessler et al. investigated a possible correlation between sHLA-G concentration in blood and HLA-G expression in thyroid cancer cells. They could not find a correlation between the surface expression and the concentration in the blood [27]. They reason that there are different induction factors for sHLA-G expression and release, as well as environmental factors that influence sHLA-G access to the blood, such as stromal thickness and vascularization. Thus, not all cytokines that induce HLA expression also appear to be involved in the release of sHLA [27]. It should be noted that Kessler et al. investigated a different HLA class and tumor entity than the present study. However, a similar mechanism could be conceivable for HLA-E [28]. Other studies report that sHLA-E is released not only by tumor cells but also by immune cells and endothelial cells [17,29]. Thus, there does not appear to be a necessary correlation between surface HLA-E expression and sHLA-E concentration.

Other studies show that there are gender differences in cancer. Men have a higher risk of developing cancer than women. It is thought that this may be due in part to differences in immune response between men and women [14]. Therefore, when considering sHLA-E concentration as a systemic component in tumorigenesis, which is also understood to be an immunologic marker [29,30], it seems quite plausible that sex differences occur.

Our results provide evidence that the immune systems of women and men do not respond equally to the genesis of oral squamous cell carcinoma. Considering that men and women do not respond equally to immune checkpoint inhibitor therapy [30,31,32], attention should be paid to gender-specific immune responses when planning future therapeutic regimens.

Our study has limitations that need to be considered. The ELISA cohort is smaller than the TMA cohort and therefore less powerful. In addition, serum samples collected over a 3-year period were used for the ELISA. It is therefore possible that the quality of the older samples is reduced compared to the quality of the younger samples.

## 4. Materials and Methods

### 4.1. Patients

Immunohistochemistry (IHC) was performed on Tissue Microarrays (TMA) to detect the surface HLA-E-expression level on oral squamous carcinoma cells. The patient samples were provided by the University Hospital of Heidelberg. Oral squamous cell carcinoma (OSCC) patients received primary surgical therapy after an interdisciplinary tumor board decision. Patients in advanced stages received adjuvant radiotherapy or radiochemotherapy based on the German national OSCC treatment guideline.

The blood samples for the enzyme-linked immunosorbent assay (ELISA) were provided by the Department of Oral and Maxillofacial Surgery at Saarland University. Samples were collected between 2020 and 2023. The study included patients with OSCC and a primary surgical therapy. Exclusion criteria were prior oncological therapies or any history of malignant disease. Furthermore, blood samples of the control group were collected. Prerequisites for participation in the control group were the absence of previous illnesses like cancer, dementia, diabetes, depression, and autoimmune diseases. All participants gave written informed consent. 

Histopathological evaluation of the tumors and grading were performed by the Department of Pathology at the University Hospital Heidelberg and the Department of Pathology at the University of Saarland. Tumor classification was based on the TNM classification current at the time of assessment (7th edition for tissue samples for immunohistochemistry and 8th edition for tissue and blood samples for ELISA).

### 4.2. Immunohistochemistry

TMAs were stained according to the following protocol. For preparation, deparaffinization and hydrogenated with xylene (2×) (VWR International, Radnor, PA, USA ) and a descending ethanol (VWR International) series (Ethanol 100% 2 × 5 min, Ethanol 96% 1 × 5 min, Ethanol 75% 1 × 5 min) was performed.

To unmask the antigens, the slides were boiled in a 10 mM citrate buffer solution (pH 6.0, 40 min). Then, they cooled down to room temperature (RT) for 20 min. After the sections were washed once for 5 min in PBS (Thermo Fisher Scientific, Waltham, MA, USA), BLOXALL Endogenous Enzyme Blocking Solution (Vector Laboratories, Newark, CA, USA) was added to the slides. After an incubation of 10 min in the humidity chamber, the slides were washed once with PBS for 5 min as previously described. To avoid cross-reactions, the slides were incubated with 2.5% Normal Horse Serum (Vector Laboratories) (30 min, humid chamber). The first anti-HLA-E antibody (Anti-HLA E antibody [MEM-E/02] (ab2216)) (1:150) (Abcam, Cambridge, UK) was added to the slides and incubated overnight (4° C, humid chamber). After they were washed with PBS (2 × 5 min) and with PBS-T (1 × 5 min) (PBS with Tween 20, Thermo Fisher Scientific), the slides were incubated with the second antibody (ImmPRESS Universal Antibody, Horse Anti-mouse/Rabbit IgG; Peroxidase; Vector laboratories) (room temperature, humid chamber). After two washes with PBS (2 × 5 min), the ImmPACT DAB EqV substrates (Vector Laboratories) (50:50) were freshly prepared, mixed and added to the slides to visualize the reaction of the second antibody. After exactly 60 s, the solution was removed and the slides were washed in PBS (2 × 5 min). Hematoxylin (Merck KGaA, Darmstadt, Germany) was used for counterstaining (1–2 s). Then, the slides were washed under running tap water (10 min). Subsequently, the staining was fixed by reversing the alcohol series (first ethanol 70%, 95%, 100%, then xylene twice). Finally, the slides were coated with Eukitt (Merck KGaA) and covered with a coverslip.

The slides were scanned with Axio Scan.Z1 (Zeiss, Oberkochen, Germany) and visualized with ZEN 3.8 (Blue edition)-Software (Zeiss) for further evaluation. The analysis of the stained TMAs was performed by three observers. Each core was evaluated according to the criteria “density of the stained cells” (1–4) and “number of the stained cells” (1–4). Values of 0–3 were considered as a low density or number. A value of 4 (for density and number) was defined as a high HLA-E expression.

### 4.3. Blood Samples

Venous blood from 32 tumor patients and 42 control group participants was used. After the serum was clotted (30 min at RT), it was centrifuged at 2000× *g* for 10 min (Hermle AG, Gosheim, Germany). Then, the serum was filled into cryotubes (Sarstedt AG & Co., Nümbrecht, Germany) and stored in M. Frosty (Nalgene, Rochester, NY, USA) overnight (−80 °C). Then, the tubes were stocked in nitrogen.

### 4.4. Enzyme-Linked Immunosorbent Assay

HLA-E levels were determined using an ELISA kit (antibodies-online GmbH, Catalog No. ABIN6956585) that was used according to the manufacturer’s instructions (antibodies-online GmbH). The readout was performed by a microplate reader (Thermo Fischer Scientific) at a wavelength of 450 nm. Serum concentration of HLA-E was calculated by using a standard curve.

### 4.5. Statistical Analysis

The statistical analysis was performed using Statistical Package for the Social Sciences 28.0.1 (SPSS, Chicago, IL, USA) and GraphPad Prism Version 9.1.5 (GraphPad Software, Inc., Boston, MA, USA). For the descriptive statistic, data were documented as total numbers and percentage relating to the group. For the TMA Cohort, Spearman’s rank correlation test was used to examine relations between the variables. Survival analyses were performed by using Kaplan–Meier curves. Log-Rank tests were used to determine differences between the groups. Furthermore, potential prognostic variables were examined using multivariate Cox-regression. For the ELISA-cohort, Kolmogorov–Smirnov test was used to determine a normal distribution. Descriptive statistics and correlation analyses were performed equivalent to those of the TMA cohort. For independent pair samples, Mann–Whitney U-tests were performed. The statistical comparison between two matched pair samples was performed by using the Wilcoxon test. As usual, a *p*-value of <0.05 was considered as statistically significant.

## 5. Conclusions

In summary, we demonstrated that high HLA-E expression can be evaluated as a sex-independent marker for survival in patients with OSCC and is associated with advanced tumor stage. In contrast to membrane HLA-E expression, the fluid component of HLA-E was shown to be differentially detectable in male and female tumor patients, allowing us to conclude sex differences in the immunological response against tumor tissue. Our results indicate that sex-specific adjustment of tumor therapy should be considered for patients with OSCC.

## Figures and Tables

**Figure 1 ijms-24-16699-f001:**
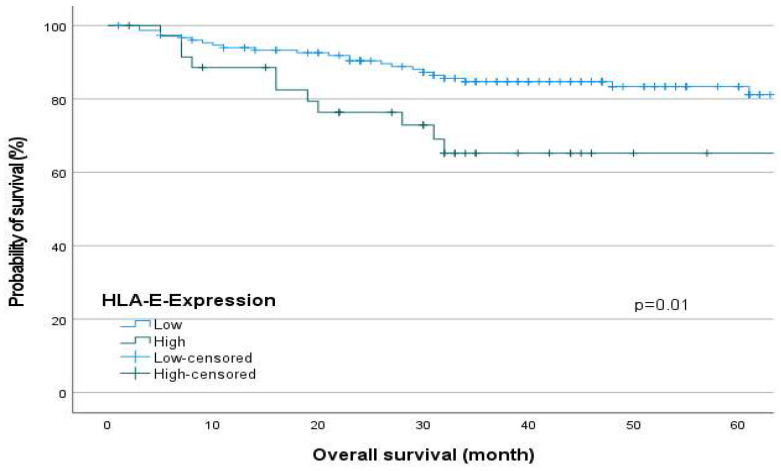
The Kaplan–Meier curve shows overall survival of OSCC patients depending on HLA-E expression. Patients with high HLA-E expression in primary tumors showed worse overall survival (Log-Rank test, *p* = 0.01).

**Figure 2 ijms-24-16699-f002:**
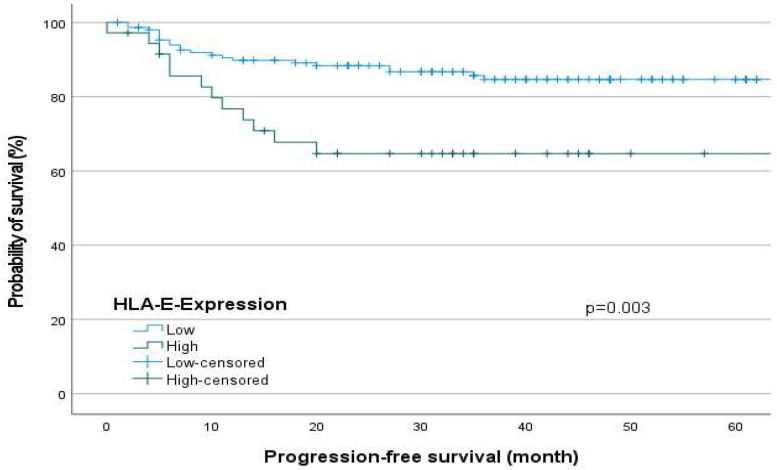
The Kaplan–Meier curve shows progression-free survival of OSCC patients depending on HLA-E expression. Patients with high HLA-E expression in primary tumors showed worse progression-free survival (Log-Rank test, *p* = 0.003).

**Figure 3 ijms-24-16699-f003:**
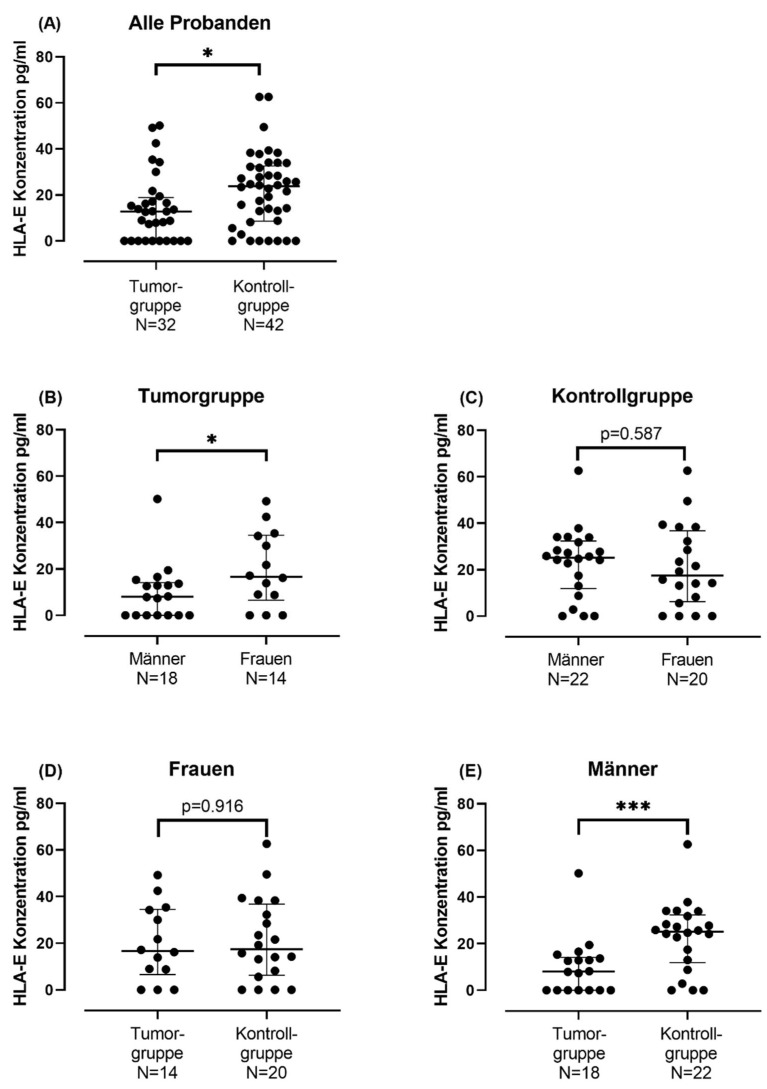
Comparison of sHLA-E concentrations between different groups. The statistical comparison between two groups was performed using the Mann–Whitney U-test. The median is represented by a bold horizontal bar. The interquartile range is marked by the whiskers. Significant differences between the groups are indicated by asterisks. * indicates a *p*-value < 0.05. *** indicates a *p*-value < 0.001. (**A**) Scatter plot showing the differences in sHLA-E levels between the tumor and control groups. Mann–Whitney U-test showed significantly higher sHLA-E concentration in control group than in tumor group (*p* = 0.021). (**B**) The scatter plot shows the sHLA-E concentrations of women and men in the tumor group. Mann–Whitney U-test showed significantly higher sHLA-E concentration in females than in males (*p* = 0.049). (**C**) The scatter plot shows the sHLA-E concentrations of women and men in the control group. Mann–Whitney U-test showed no difference in the sHLA-E concentration between females and males of the control group (*p* = 0.587). (**D**) Scatter plot showing the differences in sHLA-E levels between the female subjects in the tumor group and the female subjects in the control group. Mann–Whitney U-test showed no differences in sHLA-E concentration in females of the tumor group compared to those of the control group (*p* = 0.916). (**E**) Scatter plot showing the differences in sHLA-E levels between the male subjects in the tumor group and the male subjects in the control group. Mann–Whitney U-test showed significantly lower sHLA-E concentration in males of the tumor group compared to those of the control group (*p* = 0.001).

**Table 1 ijms-24-16699-t001:** Overview of clinicopathological parameters of the Tissue Microarray cohort.

Characteristics	Group	Quantity (Percent)
Sex	Male	137 (61.7)
	Female	85 (38.3)
Age	<64 years	109 (49.1)
	>64 years	113 (50.9)
Localization	Oral cavity	222 (100)
Histology	Squamous cell carcinoma	222 (100)
T-classification	1	82 (37)
	2	72 (32.4)
	3	8 (3.6)
	4	60 (27)
N-classification	Negative	147 (66.2)
	Positive	75 (33.8)
UICC-Stage	I	69 (31.1)
	II	44 (19.8)
	III	23 (10.4)
	IV	86 (38.7)
Grading	1	17 (7.7)
	2	153 (68.9)
	3	46 (20.7)
	Missing	6 (2.7)
Relaps	Yes	39 (17.6)
	No	183 (82.4)

Abbreviations: UICC = Union Internationale Contre le Cancer.

**Table 2 ijms-24-16699-t002:** Spearman’s correlation of HLA-E expression with clinicopathological parameters of oral squamous cell carcinoma patients.

Characteristics		HLA-E Low	HLA-E High	*p*-Value (Rho)
Sex	Male	93 (80.9)	22 (19.1)	0.994 (0.001)
	Female	59 (80.8)	14 (19.2)	
Age	<64 years	76 (89.4)	9 (10.6)	0.007 * (0.198)
	>64 years	76 (73.8)	27 (26.2)	
Localization	Mouth base	47 (90.4)	5 (9.6)	0.007 * (0.196)
	Tongue	38 (86.4)	6 (13.6)	
	Lower jaw	43 (71.7)	17 (28.3)	
	Upper jaw	3 (75)	1 (25)	
	Lower lip	0 (0)	1 (100)	
	Buccal plane	7 (63.6)	4 (36.4)	
	Soft palate	10 (83.3)	2 (16.7)	
T-classification	1	58 (90.6)	6 (9.4)	0.003 * (0.217)
	2	52 (82.5)	11 (17.5)	
	3	4 (57.1)	3 (42.9)	
	4	38 (70.4)	16 (29.6)	
N-classification	Negative	109 (83.8)	21 (16.2)	0.119 (0.114)
	Positive	43 (74.1)	15 (25.9)	
UICC-Stage	I	49 (89.1)	6 (10.9)	0.002 * (0.220)
	II	38 (88.4)	5 (11.6)	
	III	16 (84.2)	3 (15.8)	
	IV	49 (69)	22 (31)	
Grading	1	12 (75)	4 (25)	0.647 (−0.034)
	2	102 (81)	24 (19)	
	3	33 (82.5)	7 (17.5)	
Relaps	Yes	24 (68.6)	11 (31.4)	0.041 * (0.149)
	No	128 (83.7)	25 (16.3)	

Abbreviations: UICC = Union Internationale Contre le Cancer, * = *p*-value < 0.05.

**Table 3 ijms-24-16699-t003:** Multivariate Cox-regression with sex, age, N-classification, UICC-Stage and HLA-E expression.

Characteristics		Progression-Free Survival		Overall Survival	
		*p*-Value	HR (95% CI)	*p*-Value	HR (95% CI)
Sex	Male vs. Female	0.677	0.862 (0.427–1.738)	0.83	0.925 (0.457–1.874)
Age	<64 years vs. >64 years	0.350	0.712 (0.350–1.452)	0.46	0.988 (0.956–1.021)
N-classification	Positive vs. negative	0.005 *	4.764 (1.607–14.120)	0.032 *	2.663 (1.086–6.526)
UICC-Stage		0.257	0.775 (0.498–1.205)	0.686	1.082 (0.738–1.586)
HLA-E expression	High vs. Rest	0.004 *	3.129 (1.443–6.787)	0.033 *	2.328 (1.071–5.060)

Abbreviations: UICC = Union Internationale Contre le Cancer, HR = Hazard ratio, CI = confidence range, * = *p*-value < 0.05.

**Table 4 ijms-24-16699-t004:** Overview of clinicopathologic parameters of the enzyme linked immunosorbent assay of patient group and control group.

Characteristics		Tumor Group (Percent)	Control Group (Percent)
Sex	Male	18 (56.2)	22 (52.4)
	Female	14 (43.8)	20 (47.6)
Age	<65 years	16 (50)	28 (66.7)
	>65 years	13 (40.6)	13 (31)
	Missing	3 (9.4)	1 (2.3)
HLA-E	Positive	22 (68.8)	35 (83.3)
	Negative	10 (31.3)	7 (16.7)
Histology	Squamous cell carcinoma	32 (100)	
T-classification	1	6 (18.8)	
	2	7 (21.9)	
	3	7 (21.9)	
	4	12 (37.4)	
N-classification	Negative	22 (68.8)	
	Positive	9 (28.1)	
	Missing	1 (3.1)	
UICC-Stage	I	6 (18.8)	
	II	5 (15.6)	
	III	7 (21.9)	
	IV	13 (40.6)	
	Missing	1 (3.1)	

Abbreviations: UICC = Union Internationale Contre le Cancer.

## Data Availability

Data supporting the findings of this study are available from the corresponding author upon reasonable request.

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
