# Peer review of "HLA-E and Its Soluble Form as Indicators of a Sex-Specific Immune Response in Patients with Oral Squamous Cell Carcinoma"

_ijms, 2023, doi:10.3390/ijms242316699_

Round 1

Reviewer 1 Report (Previous Reviewer 1)

Comments and Suggestions for Authors

The authors have significantly improved the MS and have incorporated my major suggestions. I have only some minor questions:

Fig. 4 - please add the number of patients/volunteers in each group either directly on a graph or in the caption.
I would suggest adding a comparison of W vs M for healthy controls as well - there seems to be a notable difference
What is the measure of central tendency for fig. 4? Measures of central tendency should be marked in Figures 3, 5 and 6.
Again I would suggest merging figures 3, 4, 5, 6 into a single figure. It would increase the clarity.

Comments on the Quality of English Language

Acceptable.

Author Response

Dear Reviewer,

Thank you very much for reviewing my manuscript again and for the advice you have given me. I hope I have been able to implement them all.

  • 4 - please add the number of patients/volunteers in each group either directly on a graph or in the caption.

Many thanks for the tip. The number of participants for each group has been added to the graph.

  • I would suggest adding a comparison of W vs M for healthy controls as well - there seems to be a notable difference
    The sHLA-E concentrations of the male and female subjects in the healthy control group were analyzed for statistically significant differences using a Mann-Whitney U test. No statistically significant difference was found between the groups.

  • What is the measure of central tendency for fig. 4? Measures of central tendency should be marked in Figures 3, 5 and 6.
    The median and IQR were provided for each graph in Figure 3. I decided to use the median because there is no normal distribution for the sHLA-E concentration in this collective.

  • Again I would suggest merging figures 3, 4, 5, 6 into a single figure. It would increase the clarity

Please excuse the fact that you had to make this comment again. All graphs are now combined into one figure (Figure 3). I hope that I have now been able to implement your suggestion.

Reviewer 2 Report (Previous Reviewer 2)

Comments and Suggestions for Authors I checked the revised version of the manuscript   entitled "HLA-E and its soluble form as indicators of a sex-specific immune response in patients with Oral Squamous Cell Carcinoma". I believe the manuscript has been significantly improved and now warrants publication in ijms.  

Author Response

Dear Reviewer,

Thank you very much for your feedback. I have made a few changes to the graphical representations in the ELISA results section.

This manuscript is a resubmission of an earlier submission. The following is a list of the peer review reports and author responses from that submission.

Round 1

Reviewer 1 Report

Comments and Suggestions for Authors

Major:
1. Please provide rho values for every corelation
2. An additional cohort for sHLA-E with larger sample size, especially for control group, is crucial and needed. Current sample size of control group is unacceptable.
3. Line 186. "No significant sex depending difference in sHLA-E concentration could be found in the control group" - is this a joke? A comparison of 2 vs 1?!
4. Please provide all graphs for sHLA-E as a single figure. Please provide individual values on graphs e.g. dot-plot style graphs.
5. Figure 2 - mean? It was previously stated that data distribution is not normal... Please use median, IQR instead.
6. Line 202 - the difference is not significant. Statisticial difference can be eihter significant or not, there is nothing in-between.

Minor
1. Lines 36-37 and similar: please change . into , - the proper separation of thousands.
2. Line 156: lymphnotes
3. Please provide cat number for ELISA kit
4. Please provide clone and manufacturere of HLa-E antibody

Comments on the Quality of English Language

Language acceptable. Please double check for typos.

Reviewer 2 Report

Comments and Suggestions for Authors

The study of Anne Radermacher  and Colleagues analyzed HLA-E immunohistochemical expression retrospectively in a cohort of oral squamous cell carcinomas (OSCC) patients after surgical therapy and the impact on tumor progression and survival (n=222). In a second prospective cohort, the concentration of soluble (s)HLA-E in the serum of 19 patients with OSCC and a healthy control group (n=4) was investigated.

In addition, Authors focused on sex differences in preoperative and postoperative sHLA-E levels of men and women with OSCC. Finally, they analyzed the evolution of sHLA-E levels over time after oncologic surgery and discussed the results in relation to a sex-specific application of immunotherapeutics

Major issue: In the prospective study the number of OCCS patients is too low. Moreover I am perplexed that only 4 healthy donors could be enrolled in the study and results of only 3 are shown.

I am not confident that the results support the conclusion of the Authors “Our results indicate that sex-specific adjustment of tumor therapy should be considered for patients with OSCC.”

 1.      Abstract line 27:” We demonstrated that increased HLA-E expression promotes tumor  progression regardless of sex. However, sHLA-E levels differ in male and female OSCC patients.  This effect appears to prevail after removal of the tumor mass, suggesting a difference in the sex-specific interindividual immune response.”But after removal of tumor mass there is no sex difference in the values of sHLA-E. See supplementary Table S5.

2.      In the retrospectively study, Table 2 show that there is no significative difference between male and female in the expression of HLA-E, high or low. There is a correlation only with age, independently from the sex. The same for Table 3, that  show Multivariate Cox-regression analysis with sex, age, N-classification, UICC-Stage and HLA-E expression. Again no differences between male and female for Progression free survival and Overall survival, but only for tissue expression of high or low HLA-E.

 3.      For the prospective study Authors evaluated serum HLA-E concentration in a cohort of 19 OSCC patient and 3 healthy control.  In Table S4, however OCCS patients are only 18 and control donors are only 3. Why?

 4.      Fig 2 In the graph it must be indicated the bar of standard deviation or standard error. It represent values of 18 patients and 3 Healthy donors. The same should be done in fig. 4

 5.      In Table S5 levels of sHLA-E before and 1 week after surgery of only 10 OCCS patients  are evaluated, four men and six female. Why? Moreover there was no statistically significant difference between the HLA-E serum level on surgery day and one week after, related to sex (Men p=0.25;  Women p=0.313) (Supplemental table 5).

6.      Line 182 “Female OSCC patients showed higher sHLA-E concentration prior to cancer surgery” But in Fig.5 It seems that the level of sHLA-E is increased in the female one week post surgery in comparison to levels before surgery, as shown in fig. 3. Please, explain.

 7.      References section The name of many journals are written sometime in extenso and sometime are abbreviated. You must standardize.